# A Preliminary Evaluation of the Comparative Efficacy of Gel-Based and Oil-Based CBD on Hematologic and Biochemical Responses in Dogs

**DOI:** 10.3390/vetsci12040342

**Published:** 2025-04-07

**Authors:** Wassana Puttharaksa, Rangsun Charoensook, Rongdej Tungtrakanpoung, Niramon Hoidokhom, Saowaluk Rungchang, Bertram Brenig, Sonthaya Numthuam

**Affiliations:** 1Division of Animal Science and Feed Technology, Department of Agricultural Sciences, Faculty of Agriculture, Natural Resources and Environment, Naresuan University, Phitsanulok 65000, Thailand; wassanap66@nu.ac.th (W.P.); rangsunc@nu.ac.th (R.C.); niramonh67@nu.ac.th (N.H.); 2Department of Biology, Faculty of Science, Naresuan University, Phitsanulok 65000, Thailand; rongdejt@nu.ac.th; 3Department of Agro-Industry, Faculty of Agriculture, Natural Resources and Environment, Naresuan University, Phitsanulok 65000, Thailand; saowalukr@nu.ac.th; 4Institute of Veterinary Medicine, University of Göttingen, 37077 Göttingen, Germany; bbrenig@gwdg.de

**Keywords:** CBD, cannabidiol, dogs, stress, cortisol, hematology, biochemical response

## Abstract

Cannabidiol (CBD) is increasingly used in dogs to alleviate stress, pain, and inflammation. However, oral administration has limitations due to poor bioavailability and inconsistent absorption. This study examined the effects of daily oral supplementation with CBD gel, CBD oil, and a placebo in shelter dogs subjected to acute stress. Both CBD formulations were apparently well tolerated, with no adverse hematological or biochemical effects. CBD-treated dogs exhibited lower post-stress cortisol levels compared to controls, with the gel formulation showing a pattern toward greater attenuation. Principal component analysis (PCA) indicated that CBD supplementation was associated with distinct shifts in blood profiles, particularly in lymphocyte percentages and IgG levels. These findings suggest that gel-based CBD may offer a promising strategy for stress management in dogs, warranting further investigation.

## 1. Introduction

Dogs are frequently exposed to a variety of stress-inducing situations, including loud noises (e.g., thunderstorms and fireworks), travel, separation, and unfamiliar environments. Such stressors trigger not only behavioral changes, such as panting, licking, and vocalization, but also measurable physiological and biochemical responses, notably shifts in cortisol levels, immune modulation, and metabolic disturbances [1]. Prolonged exposure to stress may contribute to chronic health challenges, including immune suppression, inflammatory dysregulation, and increased susceptibility to disease [2,3,4].

In recent years, cannabidiol (CBD), a non-psychoactive compound derived from *Cannabis sativa*, has gained significant attention in veterinary medicine due to its potential therapeutic effects in companion animals. Reports suggest that CBD may offer anti-inflammatory, analgesic, and anxiolytic benefits, making it a promising adjunct therapy for managing chronic pain, anxiety, and stress-related conditions in dogs [5,6,7,8,9]. This growing interest has led to the development of various CBD formulations, including oils, capsules, treats, and gels, each claiming to offer distinct advantages in terms of bioavailability and ease of administration.

A variety of CBD formulations are now available for dogs, with oil-based preparations being the most common. However, the hydrophobic property of CBD poses considerable challenges for oral administration, limiting its bioavailability due to poor water solubility, inconsistent absorption, and extensive first-pass metabolism [10,11]. In response to these challenges, alternative delivery systems such as emulsions and semi-solid formulations have been developed to improve absorption and enhance systemic exposure in both human and veterinary applications [11,12,13,14,15]. Oleogels, semi-solid systems formed by immobilizing oil within a gel matrix, have demonstrated potential as carriers for lipophilic bioactive compounds, including CBD. Studies indicate that oleogel systems not only enhance bioavailability but may also allow for more controlled and sustained release of the active ingredient [16,17,18]. These characteristics suggest that CBD-infused oleogels may improve the consistency and reliability of therapeutic effects compared to conventional oil-based forms.

Despite growing interest in CBD as a therapeutic option for companion animals, there is limited research on its long-term effects on hematologic and biochemical markers, particularly in dogs experiencing chronic stress. Most existing work has focused on pharmacokinetics and behavioral outcomes [10,13,19], with limited attention to the question of how sustained CBD administration influences blood health markers and systemic biochemical balance. These blood-based biomarkers, including hematology profiles, liver function indicators, and stress-related hormones like cortisol, offer valuable insights into how well CBD is tolerated, whether it supports immune-metabolic resilience, and whether different delivery systems influence these responses differently over time.

This study investigated dogs from animal shelters, a population known to experience chronic stress due to suboptimal living conditions, inconsistent nutrition, and limited veterinary care. Such factors not only result in compromised immune and metabolic function in these dogs but also make them ideal candidates for evaluating the potential health benefits of CBD supplementation. This preliminary study therefore aimed to investigate the effects of a gel-based CBD formulation on hematologic and biochemical responses in shelter dogs. By comparing its efficacy to that of conventional CBD oil, the present study evaluated whether CBD-infused oleogels provided more consistent or enhanced effects on key blood health parameters. The findings from this study may contribute to the optimization of CBD administration in veterinary medicine and provide valuable insights into the development of more effective and consistent CBD-based strategies for supporting metabolic and immune health in dogs.

## 2. Materials and Methods

### 2.1. Animals and Housing

Twelve adult mixed-breed dogs (six males and six females) of unknown ages, with an average body weight of 16 ± 1.4 kg, were recruited from a private shelter registered under the Department of Livestock Development in Phitsanulok, Thailand; this study was conducted on-site at the shelter for 14 days. The study protocol was reviewed and approved by the Naresuan University Animal Care and Use Committee (NUACUC; reference no. 68 01 006), ensuring compliance with applicable animal care standards and ethical research guidelines.

All participants were selected based on their body condition score (BCS) between 3 and 4 out of 9, indicating a thin body condition and potential malnutrition risk [20], and absence of severe clinical illness, defined as persistent diarrhea, inappetence, lethargy, signs of pain, or visible signs consistent with neoplastic disease, as confirmed through veterinary examination prior to enrollment. In addition, all dogs exhibited observable stress-related behaviors when exposed to unfamiliar humans, including panting (rapid, open-mouth breathing unrelated to heat dissipation), cowering (lowering the body posture or crouching close to the ground), and retreating (actively moving away from the unfamiliar person) [2,21].

The dogs had typically lived with free access to both indoor and outdoor areas within the shelter. During the 14-day acclimation period preceding the trial, all dogs were fed a standard commercial dry dog food containing at least 26.0% crude protein, 16.0% crude fat, and no more than 3.0% crude fiber. Feeding was conducted once daily at 17:00, followed by an overnight fast before dosing at 08:00 on the following day, to ensure controlled gastrointestinal conditions for compound absorption.

### 2.2. Study Design and Treatment

This study followed a randomized, placebo-controlled, blind design conducted over a 14-day period. The twelve dogs were randomly assigned to one of three treatment groups, with four dogs per group (two males and two females): (i) CON group, receiving a placebo gel without CBD; (ii) CBDO group, receiving a CBD-predominant oil at a dose of 4 mg/kg body weight; or (iii) CBDG group, receiving a CBD gel formulation at a dose of 4 mg/kg body weight.

The CBD gel formulation was prepared as an oleogel, composed of medium-chain triglycerides (MCT oil) as the carrier, structured using Cab-O-Sil (fumed silica) to form a semi-solid gel matrix. For the CBDO and CBDG groups, Full Spectrum CBD Oil 40% was incorporated into the formulation at the target dose of 4 mg/kg body weight. This CBD oil was sourced from Salus Bioceutical (Thailand) Co., Ltd., Chiang Mai, Thailand. It has been registered as an herbal health product under the Thai Traditional and Alternative Medicine Department (Registration No. G 1/68). The placebo oleogel followed the same formulation but excluded CBD. The oleogel preparation was modified from the method described by Wróblewska et al. (2020) [22] to optimize the formulation for this study.

All treatments were administered once daily at 08:00 following the overnight fast. Each treatment was embedded within a meatball prepared from commercially available ground meat, serving as an oral delivery vehicle to ensure complete ingestion. Each dog was individually observed to confirm that the entire meatball, including the embedded treatment, had been fully ingested.

### 2.3. Stress Induction

Solitary confinement stress was selected as the stressor in this study based on the dogs’ background, as they had lived with free access to both indoor and outdoor areas and had no prior experience of individual confinement. Each dog underwent stress induction on days 1, 7, and 14 to evaluate the effects of CBD under both resting and stress-induced conditions. The experimental timeline, including treatment administration, baseline sampling, stress induction, and post-test procedures, is illustrated in Figure 1.

On days 1, 7, and 14, prior to the treatment administrations, all dogs were weighed, had their rectal temperature recorded, and underwent a pre-treatment veterinary examination. Pre-treatment blood sampling was then performed to establish baseline cortisol levels and hematological parameters (Hour 0). After that, at 08:00–08:30, the assigned treatment was administered by shelter staff. Immediately after treatment administration, each dog was placed individually into a confinement cage sized 118 × 79 × 100 cm, isolated from both visual and physical contact with other dogs, for a period of 2 h. This controlled isolation served as the acute stressor for the study. At the end of the 2-h confinement period (Hour 2), post-stress blood samples were collected, and the dogs were released to roam freely within the shelter.

### 2.4. Blood Collections and Analyses

On days 1, 7, and 14, blood samples were collected at Hour 0 and Hour 2. At Hour 0, immediately before treatment administration, a 5 mL blood sample was collected via cephalic venipuncture using a 22G intravenous catheter (NIPRO, Osaka, Japan). This sample was divided into a single red-top clot activator tube, stored on ice, and centrifuged at 3000× *g* for 10 min to separate serum. Serum cortisol was measured using the Architect Ci4100 analyzer (Abbott Diagnostics, Chicago, IL, USA) based on the Chemiluminescent Microparticle Immunoassay (CMIA) technique. Immunoglobulin concentrations were analyzed using the Atellica NEPH 630 System (Siemens Healthineers, Erlangen, Germany) via Nephelometry. Additional serum chemistry parameters—ALP, ALT, AST, albumin, globulin, total protein, and BUN—were analyzed using the PKL Auto Chemistry Analyzer (PKL s.r.l., Rome, Italy), which applies photometric analysis with reagent kits from BioSystems S.A. (Barcelona, Spain).

At Hour 2, immediately after the 2-h solitary confinement period, a second 5 mL blood sample was collected from the same catheter, which had been maintained in situ since the initial sampling. This second sample was split into a red-top clot activator tube for repeated cortisol analysis and an EDTA tube for hematological analysis. Hematological parameters included white blood cell count (WBC), red blood cell count (RBC), hemoglobin (Hb), hematocrit (HCT), mean corpuscular volume (MCV), mean corpuscular hemoglobin (MCH), mean corpuscular hemoglobin concentration (MCHC), platelet count, and a differential white blood cell count (neutrophils, lymphocytes, monocytes, eosinophils). All blood collection and handling were performed under aseptic conditions by a licensed veterinarian to ensure sample integrity and safeguard animal welfare. All blood sample analyses were performed at Medilab Medical Technology Clinic, Thailand (License No. 65107000262).

### 2.5. Statistical Analysis

All statistical analyses were performed using Python version 3.11. Data normality was assessed using the Shapiro-Wilk test from the scipy.stats library (v1.10), which confirmed that all parameters met the assumption of normal distribution. For hematological, biochemical, immunoglobulin, and liver enzyme parameters, two-way repeated measures ANOVA was conducted using the pingouin library (v0.5.3) to assess the main effects of treatment group (CON, CBDO, CBDG), time (Day 1, Day 7, Day 14), and their interaction.

For serum cortisol, which was measured twice daily (pre-stress at Hour 0 and post-stress at Hour 2), separate two-way repeated measures ANOVA was performed for each timepoint to evaluate the effects of treatment group and sampling day, as well as their interaction. Whenever significant main effects or interactions were detected, Bonferroni-adjusted post-hoc comparisons were applied to control for multiple comparisons. All results for normally distributed data are reported as mean ± standard deviation (SD).

To evaluate multivariate patterns across hematological and biochemical parameters, Principal Component Analysis (PCA) was performed using combined hematological and biochemical data from all 36 blood samples collected from the 12 dogs across the three sampling days (days 1, 7, and 14). Prior to PCA, all parameters were normalized using z-score normalization, which standardizes each variable to have a mean of 0 and a standard deviation of 1, to ensure that differences in scale between variables did not unduly influence the analysis. PCA was conducted to explore the overall pattern of blood parameter variation across treatment groups and time points and to evaluate potential clustering patterns related to CBD administration. Linear Support Vector Machine (SVM) classifiers were applied to the principal components to visualize group separation boundaries within the PCA plot. All multivariate analyses were performed using scikit-learn (v1.3), and data visualization was carried out using Matplotlib (v3.7.1) in Python.

## 3. Results

### 3.1. Hematological and Biochemical Parameters

Most hematological and biochemical parameters remained within the reference ranges for healthy dogs across all treatment groups throughout the 14-day study period (Table 1 and Table 2). No statistically significant differences were detected between groups for total protein, albumin, globulin, liver enzymes (AST, ALT, ALP), or key hematological indices including WBC count, RBC count, and platelet count (*p* > 0.05).

Throughout the study period, hemoglobin (Hb), hematocrit (HCT), and platelet counts in all groups were consistently lower than the standard reference ranges for healthy dogs [23]. This pattern was observed at Day 1 and persisted across all sampling days, reflecting the suboptimal baseline health status commonly reported in shelter dog populations [24]. However, these values did not differ significantly between treatment groups (*p* > 0.05). Despite the low baseline levels, no progressive decline or treatment-related hematologic abnormalities were observed, supporting the overall hematological safety of CBD supplementation at the tested dose.

### 3.2. Serum Cortisol and Stress Response

On Days 1, 7, and 14, all dogs underwent a 2-h solitary confinement stress test following morning treatment administration. The 2-h interval between treatment and stress induction was selected based on pharmacokinetic data from previous studies, which reported peak plasma CBD concentrations occurring approximately 1.5 to 2 h after oral administration in dogs [5,6,25]. Serum cortisol concentrations were measured at baseline (Hour 0), immediately before stress induction, and at post-stress (Hour 2), immediately after the confinement period (Table 2, Figure 2).

By Day 7 and Day 14, post-stress cortisol levels were significantly lower in both CBD-treated groups compared to controls (*p* < 0.05). This suggests that CBD supplementation may mitigate acute stress responses, with the gel formulation showing a tendency toward greater attenuation. Analysis of ΔCortisol (the difference between post-stress and baseline cortisol levels) further supported this effect (Figure 3). On both days, ΔCortisol was significantly lower in the CBD gel group compared to the control group (*p* < 0.05), with the CBD oil group also showing numerically lower ΔCortisol values, though the difference between the oil and gel groups was not statistically significant. These results indicate that both CBD formulations were effective in attenuating acute stress responses, with the gel formulation demonstrating a tendency toward greater cortisol reduction over time.

### 3.3. Immunological Responses

Serum immunoglobulin G (IgG) and immunoglobulin M (IgM) concentrations remained stable over the 14-day study period, with no statistically significant differences detected between treatment groups at any time point (*p* > 0.05) (Table 2). Regarding white blood cell subpopulations, neutrophils, monocytes, and eosinophils did not show statistically significant differences between treatment groups across all sampling days (*p* > 0.05) (Table 1). In contrast, lymphocyte percentages showed a numerical increase in both CBD-treated groups over time, with the CBD gel group exhibiting significantly higher lymphocyte percentages compared to the control group on Day 14 (*p* < 0.05). No significant differences in lymphocyte counts were observed between the CBD oil and CBD gel groups (*p* > 0.05).

### 3.4. Principal Component Analysis and Blood Parameter Associations

To explore overall trends and treatment-associated variation in blood profiles, Principal Component Analysis (PCA) was applied to summarize the combined variation across hematological and biochemical parameters using data from all 36 blood samples (12 dogs × 3 sampling days). Unlike univariate analysis, as presented in Table 1 and Table 2, which evaluates individual parameters separately across groups and timepoints, PCA was employed, as a multivariate method, to examine integrated variation across all hematological and biochemical variables collectively. This multivariate approach allows for the detection of overall trends and covariation patterns that may not be apparent through individual parameter comparisons.

The PCA score plot (Figure 4), based on PC1 and PC3, demonstrated clear separation between treatment groups. Samples from placebo-treated dogs clustered primarily in the upper half of the plot, while CBD-treated dogs (both CBD oil and CBD gel) clustered predominantly in the lower half, with CBD gel-treated dogs positioned toward the lower central region. The separation between CON and CBD groups occurred primarily along PC3, indicating that this component captured variance associated with CBD supplementation.

The accompanying PCA loadings plot in Figure 4 indicated that dogs in the CON group showed higher associations with neutrophil counts, post-stress cortisol, and ΔCortisol, while the CBD oil group aligned more closely with albumin, ALP, and WBC counts. Dogs in the CBD gel group were associated with BUN, IgG, and lymphocytes, highlighting potential immune-modulating and renal-related influences associated with the gel formulation. These treatment-related shifts were further supported by the Support Vector Machine (SVM) classification, which successfully separated the groups based on the principal component scores.

## 4. Discussion

This study evaluated the effect of daily oral administration of CBD oil and CBD gel, each at a daily dose of 4 mg/kg body weight, over a 14-day period on hematological and biochemical parameters, including cortisol, in shelter dogs exposed to acute confinement stress. The selected dose was based on previous studies that demonstrated the safety and therapeutic potential of oral CBD in dogs within the 2–4 mg/kg/day range [23,26]. Based on this evidence, 4 mg/kg was selected as a representative dose for evaluating both safety and biological effects, particularly under chronic stress conditions relevant to the shelter population. Overall, most hematological and biochemical parameters remained within reference ranges for dogs [23] and did not differ significantly between treatment groups or across sampling days (Table 1, Table 2). These findings support the safety profile of CBD at the tested dose, with no adverse effects observed in liver function markers (ALT, AST, ALP) or hematological indices, aligning with previous reports in healthy dogs [6,25,27,28]. Notably, Vaughn et al. (2020) [27] employed a comparable small group size (n = 4 per treatment) and reported no clinically relevant adverse effects even at higher doses up to 16.8 mg/kg, providing additional support for the safety data observed in the present study despite its limited sample size. To compensate for this limitation, repeated measures ANOVA was applied to increase the analytical sensitivity when evaluating time-dependent effects. This design is especially suitable for small animal studies, where using each subject as its own control helps minimize between-subject variability.

It is worth noting that baseline hemoglobin (Hb), hematocrit (HCT), and platelet counts were lower than the standard reference ranges for healthy dogs, a finding commonly reported in shelter populations, where suboptimal health, chronic stress, and nutritional deficiencies are prevalent [24,29,30]. However, these values remained stable throughout the study and did not differ significantly between treatment groups, indicating that CBD supplementation did not negatively impact or further deteriorate the hematological status of these dogs. Furthermore, no adverse hematological effects, such as anemia or leukopenia, were detected during the study period, reinforcing the safety of CBD supplementation in dogs with pre-existing physiological challenges.

Another interesting observation relates to lymphocyte counts, which showed a numerical increase in the CBD-treated groups by Day 14, despite the absence of statistically significant differences between treatment groups (Table 1). This upward pattern in lymphocyte counts, combined with the observation that IgG levels remained comparable across all groups (Table 2), raises the possibility that CBD supplementation may have influenced T-cell populations more than B-cell-mediated antibody production. This finding aligns with previous research showing that cannabinoids modulate immune function primarily through T-cell regulation and cytokine signaling, which may help maintain immune resilience under chronic stress conditions [31,32]. Additionally, these effects may be indirectly mediated by the stress-attenuating properties of CBD, which could help preserve lymphocyte homeostasis by mitigating cortisol-induced immunosuppression [33]. Beyond its role in stress-related immune modulation, CBD has been shown to regulate lymphocyte activity by interacting with immune signaling cascades that influence cell proliferation and apoptosis pathways, supporting its potential immunomodulatory effects [34,35].

This indirect immunomodulatory pathway is further supported by the observed reduction in cortisol levels in CBD-treated dogs in this study, reinforcing the role of CBD in attenuating stress responses. Both CBD oil and CBD gel significantly reduced post-stress cortisol levels compared to controls, demonstrating that CBD may help modulate acute stress responses induced by solitary confinement [25]. On Day 14, for instance, the mean post-stress cortisol level was 5.78 ± 1.33 µg/dL in the control group, compared to 4.98 ± 0.88 µg/dL in the CBDO group and 3.58 ± 0.21 µg/dL in the CBDG group, highlighting a greater reduction in the gel-treated dogs (Table 2). These findings are consistent with previous studies reporting that CBD can attenuate HPA axis activation, reducing stress-induced elevations in cortisol in both animals and humans [6,36,37], with emerging evidence suggesting possible benefits in reducing aggressive behavior in shelter dogs [38].

In addition to its effects on cortisol, Principal Component Analysis (PCA) provided further insight into how CBD supplementation influenced overall hematological and biochemical profiles. The PC1 vs. PC3 biplot demonstrated a clear clustering of samples based on treatment, with dogs in both CBD-treated groups separating distinctly from controls, particularly along PC3. This separation suggests that CBD supplementation—regardless of delivery form—contributed to a shift in blood profiles, potentially reflecting broader physiological adaptations to chronic supplementation in combination with stress reduction. The accompanying PCA loadings plot (Figure 4) revealed that samples from the CON group were more closely associated with neutrophil counts, post-stress cortisol, and ΔCortisol, indicating a heightened acute stress response accompanied by a stress leukogram [39,40]. In contrast, samples from the CBD oil group aligned more closely with albumin, ALP, and WBC counts, while the CBD gel group was associated with BUN, IgG, and lymphocytes, suggesting potential differences in immune modulation and renal-related responses between formulations [6,31,32,41,42]. The successful classification by Support Vector Machine (SVM) further confirms that these multivariate shifts were treatment-dependent, highlighting the value of combining multivariate analysis with machine learning classifiers to capture the comprehensive physiological footprint of CBD supplementation in stressed dogs. An interesting pattern emerges when comparing our multivariate findings with those reported by Mills et al. (2024) [43], particularly regarding ALP responses in dogs receiving CBD oil formulations. Although ALP levels in our study did not differ significantly between treatment groups in univariate analysis, they showed a clear association with the CBD oil group in PCA loadings. Mills et al. (2024) [43] reported a statistically significant increase in ALP following oral administration of a 4 mg/kg CBD-rich hemp extract in MCT oil over a 36-day period in dogs. Their findings—together with our multivariate pattern—suggest that oil-based formulations may may exert a more noticeable effect on hepatic biomarkers. In contrast, the gel-based formulation in our study showed no such ALP alignment and was instead associated with immunological markers, which may indicate a more favorable systemic profile in chronically stressed dogs. It is important to note that PCA was performed on pooled data from all dogs across the three sampling timepoints, allowing for the analysis of integrated hematological and biochemical variation rather than isolated comparisons at each timepoint. While univariate statistical tests, as reported in Table 1 and Table 2, assess differences in individual parameters across groups, PCA, as a multivariate method, summarizes covariation patterns among all measured variables. Although this approach does not directly enhance statistical power, it enables the detection of multivariate trends and group-level clustering that may not be apparent from univariate comparisons alone.

Interestingly, although both formulations provided comparable cortisol-lowering effects, the literature suggests that gel-based formulation may offer additional advantages over the oil in terms of both bioavailability enhancement and practical administration. As a semi-solid oleogel, the gel matrix enhances gastrointestinal retention and promotes lymphatic absorption of lipophilic compounds like CBD, which may contribute to more consistent systemic exposure [12,16,18,44]. This pharmacokinetic advantage could partly explain the observed pattern of lower ΔCortisol in the gel group compared to the oil group, particularly on later sampling days (Figure 3). The data from Table 2 show that on Day 7 and Day 14, both post-stress cortisol levels and ΔCortisol (the change from pre- to post-stress) were numerically lower in the CBD gel group compared to the oil group, even though the differences were not statistically significant. This pattern may suggest that the CBD-infused gel formulation provided more stable or sustained cortisol modulation over time; however, due to the small sample size, further studies are needed to validate this hypothesis. Such patterns are consistent with the proposed stress-attenuating effects of CBD, whereby chronic supplementation helps mitigate acute stress responses, potentially through modulation of the hypothalamic-pituitary-adrenal (HPA) axis and regulation of neuroendocrine signaling [36]. Similar findings have been reported in both canine and human studies investigating CBD’s anxiolytic and anti-stress properties under acute stress challenges [6,45].

In addition to its pharmacokinetic advantages, the gel formulation may also offer practical benefits, including greater ease of handling and improved compatibility with palatable delivery methods. Compared to conventional oil or tablet formulations, oleogel-based delivery systems provide better adhesion within soft food carriers and reduce the likelihood of spillage [46,47], making them particularly attractive for companion animals and other populations with swallowing difficulties, where ensuring complete oral intake is essential for treatment adherence and therapeutic success [46].

Future studies incorporating larger sample sizes and longer supplementation periods are warranted to validate these preliminary findings and explore the long-term immunological and physiological effects of CBD gel in dogs under chronic stress conditions. Expanding the scope of investigation to include functional assays, such as T-cell subset profiling or cytokine analysis, could also provide deeper mechanistic insights into the immunomodulatory actions of CBD. Additionally, research on the impact of gel-based CBD supplementation on feeding behavior would be valuable in assessing its potential role in appetite modulation and nutritional intake in stress-prone dogs.

## 5. Conclusions

This preliminary study found that daily oral supplementation of gel-based CBD and CBD oil at 4 mg/kg for 14 days was apparently well tolerated in shelter dogs, with no adverse hematological or biochemical effects. Both formulations reduced post-stress cortisol, with the CBD-infused gel showing a pattern toward greater attenuation, likely due to its enhanced bioavailability. According to PCA, gel-based CBD showed associations with lymphocyte responses, which may reflect underlying immunomodulatory effects. Future studies with larger cohorts are warranted to confirm these preliminary findings and further investigate the pharmacokinetics of gel-based CBD, its broader immunological effects, and its potential impact on feeding behavior in chronically stressed dogs, thereby supporting the development of optimized therapeutic strategies in veterinary medicine.

## Figures and Tables

**Figure 1 vetsci-12-00342-f001:**
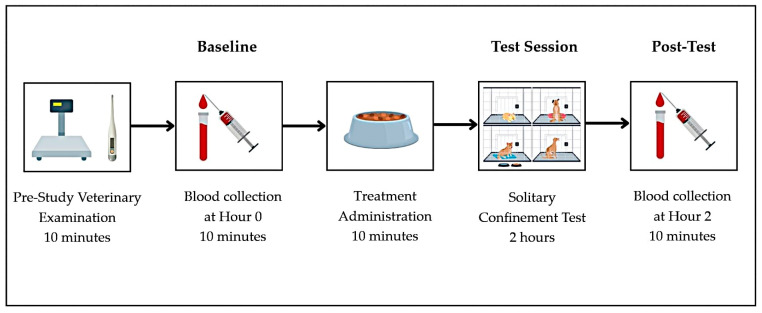
Experimental workflow illustrating test day timeline, including treatment administration, stress induction, and blood sample collections.

**Figure 2 vetsci-12-00342-f002:**
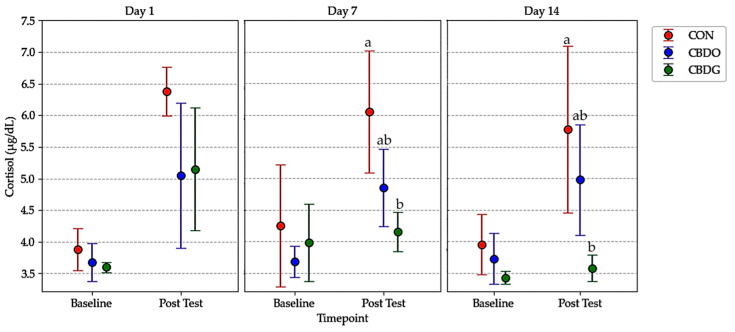
Mean serum cortisol levels (µg/dL) at baseline and post-test in dogs from the control group (CON), CBD oil group (CBDO), and CBD gel group (CBDG) at Day 1, Day 7, and Day 14. Data are presented as mean ± standard deviation (SD). Significant differences between groups are indicated by different superscript letters (*p* < 0.05), based on repeated measures ANOVA followed by Bonferroni post-hoc test.

**Figure 3 vetsci-12-00342-f003:**
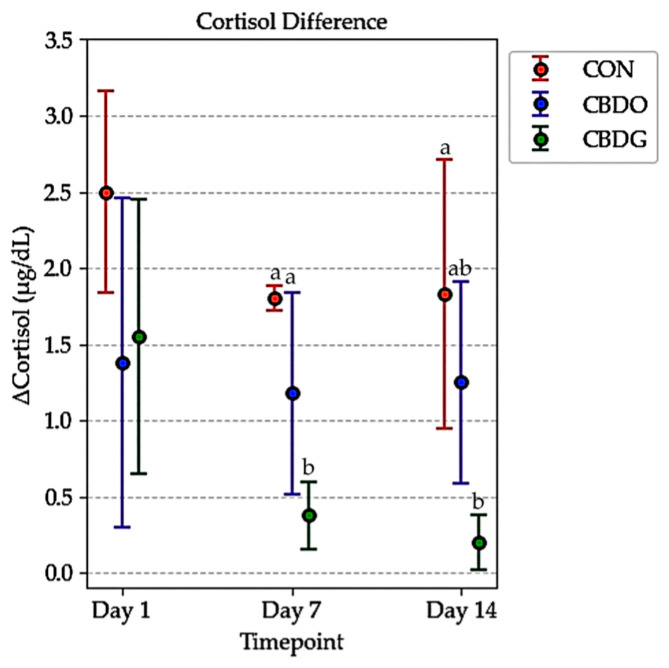
Changes in serum cortisol levels (ΔCortisol, µg/dL) in dogs from the control group (CON), CBD oil group (CBDO), and CBD gel group (CBDG) at Day 1, Day 7, and Day 14. Data are presented as mean ± standard deviation (SD). Significant differences between groups are indicated by different superscript letters (*p* < 0.05), based on repeated measures ANOVA followed by Bonferroni post-hoc test.

**Figure 4 vetsci-12-00342-f004:**
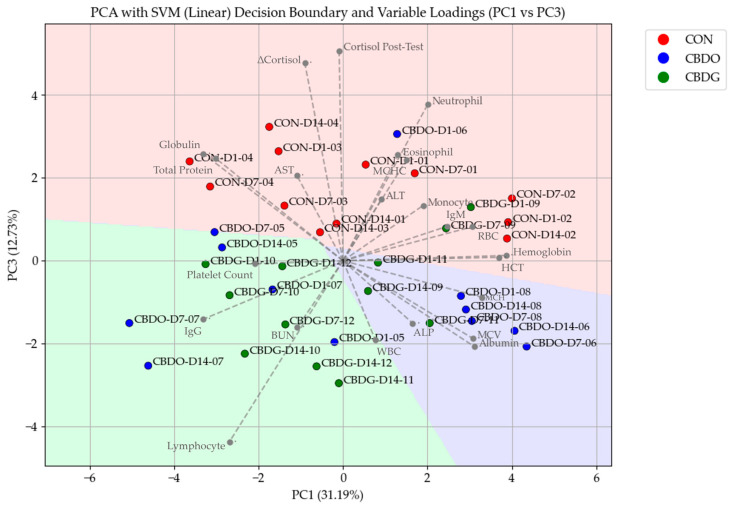
Principal Component Analysis (PCA) plot with Support Vector Machine (SVM) decision boundaries and variable loadings, comparing treatment groups (CON, CBDO, CBDG) across different time points (Day 1, 7, and 14). Data points are labeled as [Group]-[Day]-[Dog ID], representing repeated measurements of individual dogs over the study period. The variable loadings indicate the contribution of each hematological and biochemical parameter to the principal components.

**Table 1 vetsci-12-00342-t001:** Hematological parameters (Mean ± SD) in dogs from the control group (CON), CBD oil group (CBDO), and CBD gel group (CBDG) at Day 1, Day 7, and Day 14.

Parameter (Ref. Range [23])	CON			CBDO			CBDG			*p* (G)	*p* (T)	*p* (Int.)
	Day 1	Day 7	Day 14	Day 1	Day 7	Day 14	Day 1	Day 7	Day 14			
WBC (4.0–15.5 × 10^3^/mm^3^)	9.50 ± 3.21	10.78 ± 3.37	12.40 ± 5.96	13.83 ± 4.94	16.25 ± 6.16	17.33 ± 6.40	12.73 ± 4.97	16.00 ± 9.75	12.20 ± 6.81	0.502	0.093	0.251
RBC (4.8–9.3 ×10^6^/mm^3^)	4.50 ± 0.28	4.98 ± 0.53	5.10 ± 0.18	4.40 ± 0.63	4.88 ± 1.09	4.68 ± 0.95	4.70 ± 1.07	4.90 ± 1.27	4.78 ± 0.56	0.922	0.076	0.786
Hemoglobin (12.1–20.3 g/dL)	9.45 ± 1.56	10.18 ± 1.49	10.43 ± 1.16	9.50 ± 1.98	10.30 ± 3.19	10.00 ± 2.86	9.40 ± 2.13	9.95 ± 2.70	9.58 ± 0.76	0.961	0.207	0.929
Hct (36–60%)	25.00 ± 4.08	28.25 ± 3.77	29.50 ± 2.52	25.00 ± 5.35	28.75 ± 9.00	28.25 ± 8.26	26.50 ± 5.51	27.25 ± 6.99	26.75 ± 2.87	0.980	0.050	0.590
Platelets count (170–400 × 10^3^/mm^3^)	129.75 ± 47.83	75.00 ± 19.11	141.00 ± 90.06	111.00 ± 66.17	132.75 ± 76.14	105.50 ± 38.92	95.25 ± 43.25	96.00 ± 48.03	67.00 ± 35.67	0.538	0.833	0.205
MCV (58–79 fl)	55.00 ± 6.38	56.25 ± 6.34	55.50 ± 6.03	56.50 ± 4.65	58.25 ± 5.50	57.75 ± 5.56	55.75 ± 1.71	56.00 ± 1.63	56.50 ± 1.29	0.844	0.053	0.124
MCH (19–28 pg)	21.25 ± 2.63	20.50 ± 2.52	20.25 ± 2.63	21.50 ± 1.73	20.75 ± 2.06	21.00 ± 1.83	20.00 ± 0.00	20.25 ± 0.50	20.25 ± 0.96	0.787	0.058	0.056
MCHC (30–38 g/dL)	38.00 ± 0.00	36.50 ± 0.58	39.75 ± 0.50	38.00 ± 0.00	35.75 ± 0.50	36.00 ± 0.00	36.00 ± 1.15	36.75 ± 0.96	35.75 ± 0.96	0.114	0.051	0.101
Neutrophils (58–85%)	62.75 ± 9.18	62.25 ± 14.22	63.00 ± 5.72	49.50 ± 11.90	46.00 ± 5.89	49.50 ± 6.81	49.00 ± 16.27	54.50 ± 9.11	46.75 ± 3.59	0.085	0.900	0.443
Lymphocytes (8–21%)	29.75 ± 10.90	28.00 ± 12.52	28.50 ± 5.80 ^a^	32.25 ± 6.65 ^A^	37.25 ± 10.05 ^AB^	43.75 ± 5.54 ^bB^	31.50 ± 9.95 ^A^	36.75 ± 9.91 ^B^	46.75 ± 6.55 ^bB^	0.016	0.011	0.027
Monocytes (2–10%)	6.00 ± 1.63	5.75 ± 1.71	5.50 ± 1.29	6.00 ± 2.45	6.25 ± 3.30	4.50 ± 2.08	5.75 ± 0.50	6.50 ± 2.38	4.75 ± 2.87	0.987	0.285	0.927
Eosinophils (0–9%)	1.50 ± 1.00	3.75 ± 1.71	3.00 ± 0.82	2.25 ± 2.22	0.50 ± 0.58	2.25 ± 2.06	3.75 ± 5.56	2.25 ± 1.50	1.75 ± 0.50	0.602	0.922	0.212

*p* (G) represents the *p*-value for the effect of the group. *p* (T) represents the *p*-value for the effect of time. *p* (Int.) represents the *p*-value for the interaction between group and time. Data are presented as mean ± standard deviation (SD). Statistical analysis was performed using two-way repeated measures ANOVA, to assess the effects of treatment group, time, and their interaction. Bonferroni post-hoc test was applied for pairwise comparison where appropriate [23]. The mean difference is significant at the 0.05 level. Multiple comparisons were adjusted using the Bonferroni correction. Means with different superscripts ^a,b^ within the same row indicate significant differences among treatment groups at the same time point (*p* < 0.05). Means with different superscripts ^A,B^ within the same row indicate significant differences among time points within the same treatment group (*p* < 0.05).

**Table 2 vetsci-12-00342-t002:** Serum chemistry, liver enzyme, immunoglobulin, and cortisol parameters (Mean ± SD) in dogs from the control group (CON), CBD oil group (CBDO), and CBD gel group (CBDG) at Day 1, Day 7, and Day 14.

Parameter (Ref. Range [23])	CON			CBDO			CBDG			*p* (G)	*p* (T)	*p* (Int.)
	Day 1	Day 7	Day 14	Day 1	Day 7	Day 14	Day 1	Day 7	Day 14			
**Serum Chemistry**												
BUN (6–25 mg/dL)	12.75 ± 4.27	9.25 ± 2.06	13.25 ± 4.35	14.75 ± 7.85	17.50 ± 4.80	26.50 ± 13.67	16.25 ± 3.86	10.75 ± 3.86	13.00 ± 1.63	0.053	0.148	0.198
Total protein (5.0–7.4 g/dL)	11.48 ± 1.72	11.08 ± 1.26	11.43 ± 1.49	10.45 ± 1.01	10.90 ± 1.14	11.15 ± 0.98	10.50 ± 1.57	10.48 ± 1.02	10.50 ± 1.59	0.632	0.601	0.648
Albumin (2.7–4.4 g/dL)	2.18 ± 0.25	2.25 ± 0.26	2.25 ± 0.25	2.55 ± 0.58	2.43 ± 0.75	2.45 ± 0.77	2.30 ± 0.24	2.20 ± 0.20	2.15 ± 0.24	0.336	0.836	0.929
Globulin (1.6–3.6 g/dL)	9.30 ± 1.94	9.18 ± 1.73	8.83 ± 1.45	7.90 ± 1.56	8.73 ± 1.72	8.45 ± 1.84	8.20 ± 1.79	8.30 ± 1.68	8.33 ± 1.06	0.663	0.701	0.772
**Liver Function**												
AST (15–66 U/L)	29.00 ± 5.60	38.00 ± 6.48	34.50 ± 12.12	29.75 ± 7.80	27.00 ± 4.55	28.00 ± 7.79	36.50 ± 9.98	39.75 ± 11.93	35.00 ± 13.19	0.331	0.402	0.353
ALT (12–118 U/L)	33.25 ± 14.73	44.00 ± 22.70	43.50 ± 22.84	40.75 ± 24.54	31.00 ± 7.79	33.00 ± 5.03	54.75 ± 24.66	38.25 ± 17.46	32.75 ± 13.72	0.817	0.308	0.064
ALP (5–131 U/L)	58.00 ± 31.27	63.50 ± 52.13	64.00 ± 41.67	70.00 ± 47.34	47.00 ± 7.96	54.75 ± 15.00	58.25 ± 17.71	71.50 ± 20.92	83.00 ± 22.80	0.766	0.631	0.376
**Immunoglobulin**												
IgM (100–400 mg/dL)	128.58 ± 43.00	112.68 ± 49.74	112.33 ± 50.26	133.20 ± 91.77	149.75 ± 83.73	154.75 ± 80.26	74.18 ± 11.97	68.78 ± 13.48	66.53 ± 8.27	0.200	0.897	0.233
IgG (670–1650 mg/dL)	1598.00 ± 44.99	1665.00 ± 99.02	1636.50 ± 101.09	1643.25 ± 117.00	1705.75 ± 118.65	1660.25 ± 120.25	1629.00 ± 70.82	1652.50 ± 64.69	1664.50 ± 48.84	0.848	0.002	0.218
**Serum Cortisol**												
Cortisol baseline (0.5–5.5 µg/dL)	3.88 ± 0.33	4.25 ± 0.97	3.95 ± 0.48	3.68 ± 0.30	3.68 ± 0.25	3.73 ± 0.40	3.60 ± 0.08	3.98 ± 0.61	3.43 ± 0.05	0.220	0.280	0.713
Cortisol Post-Test (5.0–17.0 µg/dL)	6.38 ± 0.39	6.05 ± 0.97 ^a^	5.78 ± 1.33 ^a^	5.05 ± 1.15	4.85 ± 0.61 ^ab^	4.98 ± 0.88 ^ab^	5.15 ± 0.97 ^A^	4.15 ± 0.31 ^bAB^	3.58 ± 0.21 ^bB^	0.001	0.042	0.615
∆Cortisol µg/dL	2.50 ± 0.66	1.80 ± 0.08 ^a^	1.83 ± 0.88 ^a^	1.38 ± 1.08	1.18 ± 0.66 ^a^	1.25 ± 0.66 ^ab^	1.55 ± 0.90 ^A^	0.38 ± 0.22 ^bB^	0.20 ± 0.18 ^bB^	0.001	0.047	0.556

*p* (G) represents the *p*-value for the effect of the group. *p* (T) represents the *p*-value for the effect of time. *p* (Int.) represents the *p*-value for the interaction between group and time. Data are presented as mean ± standard deviation (SD). Statistical analysis was performed using two-way repeated measures ANOVA. to assess the effects of treatment group, time, and their interaction. Bonferroni post-hoc test was applied for pairwise comparison where appropriate. The mean difference is significant at the 0.05 level. Multiple comparisons were adjusted using the Bonferroni correction. Means with different superscripts ^a,b^ within the same row indicate significant differences among treatment groups at the same time point (*p* < 0.05). Means with different superscripts ^A,B^ within the same row indicate significant differences among time points within the same treatment group (*p* < 0.05).

## Data Availability

The original contributions presented in the study are included in the article, further inquiries can be directed to the corresponding author.

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
