# Peer review of "A Preliminary Evaluation of the Comparative Efficacy of Gel-Based and Oil-Based CBD on Hematologic and Biochemical Responses in Dogs"

_vetsci, 2025, doi:10.3390/vetsci12040342_

Round 1
Reviewer 1 Report
Comments and Suggestions for Authors
Abstract: The Abstract adequately presents the aim of the study but in line 23-25 the autors wrote “Principal component analysis (PCA) indicated that CBD supplementation was associated with distinct shifts in blood profiles, particularly in lymphocyte percentages and IgG levels. Whereas in results (line206-208) “No statistically significant differences were detected between groups for total protein, albumin, globulin, liver enzymes (AST, ALT, ALP), or key hematological indices including WBC count”. Serum immunoglobulin G (IgG) and immunoglobulin M (IgM) concentrations remained stable over the 14-day study period, with no statistically significant differences detected between treatment groups at any time point (p > 0.05) (line 245-247). And “No significant differences in lymphocyte counts were observed between the CBD oil and CBD gel groups” (line 252-254).
Introduction: The introduction section lacks an explanation of the correlation between stress and changes in leukocyte levels depending on the duration of the stressor.
Materials and methods: In the introduction, the authors pointed out the anti-inflammatory effect of CBD (line 54-55). Why did the authors not check the CRP level when writing about the anti-inflammatory effect of CBD?
Were there any differences in the results observed depending on the sex of the animal? (There were two females and two males in each group – line 118).
How big were the age differences between the animals used in the studies?
Depending on age, stress may have a different effect on the parameters tested.
On what basis was the CBD dose chosen and why was its effect checked every 7 days?
Why liver parameters were checked exactly 2 hours after CBD administration and only then?
Results: The results of the study are analytically presented. Tables and Figures explain the findings of the study.
Discussion: The results of study are sufficiently discussed. The only thing missing in the discussion is the selection of the CBD dose based on the analysis of other works.
Author Response
Abstract: The Abstract adequately presents the aim of the study but in line 23-25 the autors wrote “Principal component analysis (PCA) indicated that CBD supplementation was associated with distinct shifts in blood profiles, particularly in lymphocyte percentages and IgG levels. Whereas in results (line206-208) “No statistically significant differences were detected between groups for total protein, albumin, globulin, liver enzymes (AST, ALT, ALP), or key hematological indices including WBC count”. Serum immunoglobulin G (IgG) and immunoglobulin M (IgM) concentrations remained stable over the 14-day study period, with no statistically significant differences detected between treatment groups at any time point (p > 0.05) (line 245-247). And “No significant differences in lymphocyte counts were observed between the CBD oil and CBD gel groups” (line 252-254).
Response: We thank the reviewer for highlighting the non-significant results in several univariate analyses (e.g., total protein, liver enzymes, immunoglobulins), and we appreciate the opportunity to clarify the rationale behind the use of Principal Component Analysis (PCA) in this study.
While univariate statistical tests (e.g., repeated measures ANOVA) were performed independently for each parameter across treatment groups and timepoints to assess statistically significant differences, PCA was employed as a complementary multivariate technique to explore broader patterns of association and clustering among all hematological and biochemical variables taken together.
In our study, PCA was conducted using pooled data from all blood samples collected across the 3 timepoints and 12 dogs (n = 36 samples), in order to investigate general patterns in blood profile variation associated with CBD treatment. This differs from the ANOVA analyses, which compared parameter means per treatment group at each specific timepoint. Therefore, PCA serves not to test statistical significance, but rather to visualize trends, relationships, and multivariate clustering between treatment groups that may not be evident from single-parameter comparisons.The observed separation along PC3—driven by cortisol-related variables in the CON group and lymphocyte-related variables in the CBDG group—suggests a treatment-dependent multivariate shift in blood profiles. While individual parameters did not always reach statistical significance in univariate analysis, their collective behavior contributes meaningfully to group separation in the PCA space, especially in this study, PCA has a potential for the classification of the dogs with CBD-treated groups separating distinctly from controls.
We have now clarified this distinction in the revised manuscript (Section 3.4 Lines 286–290; Section 4 Lines 401–407), emphasizing the complementary role of PCA alongside univariate statistical testing.
Introduction: The introduction section lacks an explanation of the correlation between stress and changes in leukocyte levels depending on the duration of the stressor.
Response: In this study, changes in leukocyte levels were not part of the primary research question or hypotheses, and thus were not emphasized in the Introduction. The focus of the study was on evaluating the safety and potential stress-attenuating effects of CBD supplementation, particularly in terms of cortisol response and overall hematological and biochemical profiles.
However, during the analysis, some interesting patterns related to lymphocyte percentages emerged, particularly in the CBD-treated groups. These findings, although not statistically significant in all comparisons, were considered noteworthy and discussed in the context of possible immunomodulatory effects of CBD under chronic stress conditions. We have now clarified in the Discussion section that leukocyte-related observations were exploratory in nature and should be interpreted as such. Future studies will be needed to specifically investigate leukocyte dynamics in response to CBD and stress over varying durations.
Materials and methods:
In the introduction, the authors pointed out the anti-inflammatory effect of CBD (line 54-55). Why did the authors not check the CRP level when writing about the anti-inflammatory effect of CBD?
Response: The mention of the anti-inflammatory properties of CBD in the Introduction was intended to provide a general overview of the compound’s known biological effects, as reported in previous literature. This study, however, did not aim to investigate inflammation-related outcomes directly, and CRP or other inflammatory markers were not included in the scope of measured parameters. Our primary focus was on evaluating the safety of CBD supplementation and its potential effect on stress responses in shelter dogs, using cortisol and hematological/biochemical profiles as the main indicators.
Were there any differences in the results observed depending on the sex of the animal? (There were two females and two males in each group – line 118).
Response: We acknowledge the reviewer’s question regarding potential sex-related differences. While this study was not designed to investigate sex as an independent variable, we ensured equal numbers of males and females in each group (two males and two females) to maintain sex balance across treatment groups and minimize allocation bias. However, due to the small number of animals per sex (n = 2), statistical analysis for sex-based effects was not feasible. Therefore, we did not compare results by sex, and no conclusions regarding sex-specific responses can be drawn. We agree that future studies with larger sample sizes would be valuable to investigate potential sex-related responses to CBD supplementation in dogs.
How big were the age differences between the animals used in the studies? Depending on age, stress may have a different effect on the parameters tested.
Response: We appreciate the reviewer’s consideration regarding age-related variation. As the study was conducted using shelter dogs, the exact ages of the animals were not documented. However, all enrolled dogs were screened by a licensed veterinarian prior to inclusion, and only those estimated to be within the adult age range of approximately 5–10 years were selected. While we acknowledge that precise age confirmation was not feasible, efforts were made to avoid extreme age variability and exclude dogs that appeared geriatric or juvenile. Therefore, although age-related differences cannot be entirely ruled out, we attempted to minimize this potential confounding factor through veterinarian-based selection criteria.
On what basis was the CBD dose chosen and why was its effect checked every 7 days?
Response: The CBD dose of 4 mg/kg body weight per day was selected based on previous studies demonstrating its safety and efficacy in dogs. Regarding the sampling interval, we chose to assess outcomes every 7 days (Days 1, 7, and 14) to monitor both short- and medium-term effects of CBD supplementation. This interval allowed for the detection of physiological changes while minimizing handling-related stress in the animals. Weekly or biweekly evaluations have also been used in previous CBD studies involving companion animals. We have revised in Discussion Line 320-323.
Why liver parameters were checked exactly 2 hours after CBD administration and only then?
Response: We appreciate the reviewer’s observation and would like to clarify that liver enzyme parameters (ALT, AST, ALP) were not measured specifically at 2 hours post-administration. In this study, liver enzymes were measured once per sampling day, using the pre-treatment blood sample collected at Hour 0. In contrast, cortisol levels were measured at both Hour 0 and Hour 2 in order to assess acute stress responses related to the confinement test. Liver enzyme values were used to monitor general liver function over time during the 14-day supplementation period, rather than to evaluate short-term pharmacodynamic responses. We have revised the wording in the Materials and Methods section to clarify this point and avoid potential misunderstanding (Section 2.4 Lines 173–174)
Results: The results of the study are analytically presented. Tables and Figures explain the findings of the study.
Response: Thank you very much.
Discussion: The results of study are sufficiently discussed. The only thing missing in the discussion is the selection of the CBD dose based on the analysis of other works.
Response: Thank you very much. We have added the reason of selected CBD dose as your suggestion.

Reviewer 2 Report
Comments and Suggestions for Authors
General comments:
One major issue is related to the low sample size (n=4 per group). Even the authors make a mention that is a preliminary study and report other study with the sample number, there is an apparent lack of statistical power for several outcomes. This relevant issue needs to be adequately reported during the interpretation of the results and conclusions. Secondly, I´m not sure what the ethical implications are due to the use of shelter dogs for experimental research (it is not possible to obtain free consent from the patients) when active substances are administered. I will be neutral about this issue in my recommendation (will be an editorial decision). I suggest to add some references like, e.g., https://doi.org/10.1038/s41598-021-82439-2; https://doi.org/10.1016/j.heliyon.2024.e31345; https://doi.org/10.3390/ani14131863; https://doi.org/10.3389/fvets.2021.645667 also for this purpose. Some improvement is required in results (pair comparison for Figures 3 and 3) and discussion (mainly related to the interpretation of the results) sections. The conclusions and abstract should be updated according to the potential revision.
Specific comments:
L30: I suggest reordering these characteristics, i.e., “extensive first-pass metabolism” at final.
L30 Principal Component Analysis
L104: I suggest defining “severe clinical illness”.
L225 (Figure 2): It is not clear what groups were different regarding Bonferroni test.
L240 (Figure 3): It is not clear what groups were different regarding Bonferroni test.
L274: Please improve the readability of Figure 2 (avoid overlays). I cannot find IG inside.
L323-325: According to Table 1, CON group presented a lower number of lymphocytes on day 14 than the other two groups. Pattern instead of trend.
326-328: Regarding absolutes values, IgM remained stable during the 3 timepoints for all groups on days 1 and 7 but decreased about 40-50% on day 14. Your sample size doesn’t allow to make a conclusion about this; IgG remained constant (only due to a14-days period?).
L332: Also, for day 14? And for both CBD and CBO groups?
L342: You can report data about cortisol levels for both treated groups.
L363: These advantages were not studied here. So please add references or clearly report that is according to the literature.
L369-372: You reports several times the term “notably”. Due to the lack of statistical power, you compare absolute values. So, 1) you make a mention to this problem for each specific comparison (reporting that a new study is required to confirm this hypothesis); 2) do not compare absolute values, or, 3) in this case, it is possible to use the PCA outcomes. Also, you use the term “Trend” other is not a statistical tendency) instead of “pattern).
L398: Was apparently well tolerated in shelter dogs, no?
L400: It is associated with ... (according to the PCA)
L401-402: Please see comment on L274.
Comments on the Quality of English LanguageNone.
Author Response
One major issue is related to the low sample size (n=4 per group). Even the authors make a mention that is a preliminary study and report other study with the sample number, there is an apparent lack of statistical power for several outcomes. This relevant issue needs to be adequately reported during the interpretation of the results and conclusions.
Response: We appreciate the reviewer’s important observation regarding the limited sample size. As noted, this study was designed as a preliminary investigation to evaluate the feasibility, tolerability, and physiological trends associated with CBD gel and CBD oil in shelter dogs. While the results provide promising insights, we acknowledge that the small sample size limits the statistical power to detect subtle or moderate effects between groups.
To mitigate this limitation, we employed repeated measures ANOVA, which evaluates within-subject changes over time and improves sensitivity by accounting for individual variability across timepoints. By analyzing temporal changes within treatment groups, this approach helped reduce the influence of inter-individual differences and strengthened the detection of potential treatment effects despite the small group size.
In addition, Principal Component Analysis (PCA) was utilized as a complementary multivariate tool to explore integrated patterns across hematological and biochemical parameters. Although PCA does not directly increase statistical power, it provides valuable insight into group-level trends and covariation structures that may not emerge from univariate comparisons alone. This multivariate perspective enhances the interpretation of treatment-related shifts and supports the overall conclusions under the constraints of a small sample size.
We have revised to more explicitly acknowledge these analytical considerations. (see Discussion lines 332–335, lines 401-408; Conclusion lines 450–451).
Secondly, I´m not sure what the ethical implications are due to the use of shelter dogs for experimental research (it is not possible to obtain free consent from the patients) when active substances are administered. I will be neutral about this issue in my recommendation (will be an editorial decision).
Response: We thank the reviewer for raising this important ethical concern. While we acknowledge that animals cannot provide free consent in the human sense, ethical oversight for studies involving shelter animals was addressed through multiple safeguards.
First, the study protocol was reviewed and approved by the Naresuan University Animal Care and Use Committee (NUACUC; protocol no. 68 01 006), ensuring adherence to national and institutional ethical standards.
Second, written informed consent was obtained from the legal guardian and operator of the registered private shelter where the study was conducted. The shelter is licensed under the Department of Livestock Development, Thailand, and the dogs enrolled were under routine veterinary care. Only clinically stable dogs were included.
Third, the active substance used—cannabidiol (CBD)—was administered at a low dose (4 mg/kg) consistent with prior safety studies in dogs. The product used is legally registered under the Thai Traditional and Alternative Medicine Department. No adverse effects were observed throughout the study, and all procedures were performed under veterinary supervision to prioritize animal welfare.
Finally, this research aims to explore supportive strategies for stress-prone shelter animals who may not otherwise receive access to novel therapeutic care. Thus, we believe the study design, oversight, and intent align with ethical principles of beneficence, refinement, and responsible use of animals in research.
I suggest to add some references like, e.g.,
https://doi.org/10.1038/s41598-021-82439-2; https://doi.org/10.1016/j.heliyon.2024.e31345;
https://doi.org/10.3390/ani14131863; https://doi.org/10.3389/fvets.2021.645667
also for this purpose. Some improvement is required in results (pair comparison for Figures 3 and 3) and discussion (mainly related to the interpretation of the results) sections. The conclusions and abstract should be updated according to the potential revision.
Response: We sincerely thank the reviewer for suggesting relevant additional references, we have added into the manuscript in Introduction and Discussion. In particular, the study by Mills et al. (2024) provided valuable insights that helped contextualize the ALP loadings observed in the CBD oil group in our PCA analysis. Their findings regarding a statistically significant increase in ALP following oral administration of a CBD- and CBDA-rich hemp extract in MCT oil helped inform our interpretation of the PCA results. While ALP levels in our study did not differ significantly between groups in univariate analysis, the PCA loadings plot indicated that the CBD oil group aligned more closely with ALP, suggesting a treatment-related pattern at the multivariate level. The consistency of this trend with the findings from Mills et al. reinforces the possibility that oil-based CBD formulations may be associated with hepatic metabolic responses. This reference has now been cited and discussed accordingly in the revised manuscript (Discussion, lines 390–397). We have also revised several parts of the Results, Discussion and Conclusion sections to improve clarity and strengthen interpretation.
Specific comments:
L30: I suggest reordering these characteristics, i.e., “extensive first-pass metabolism” at final.
Response: Revised as suggested.
L104: I suggest defining “severe clinical illness”.
Response: Revised in Line 106-107.
L225 (Figure 2): It is not clear what groups were different regarding Bonferroni test.
Response: Revised by using different superscript letters in figure 2.
L240 (Figure 3): It is not clear what groups were different regarding Bonferroni test.
Response: Revised by using different superscript letters in figure 3.
L274: Please improve the readability of Figure 4 (avoid overlays). I cannot find IG inside. Response: Revised the layout position as suggested.
L323-325: According to Table 1, CON group presented a lower number of lymphocytes on day 14 than the other two groups. Pattern instead of trend.
Response: Revised as suggested, Line 349.
L 326-328: Regarding absolutes values, IgM remained stable during the 3 timepoints for all groups on days 1 and 7 but decreased about 40-50% on day 14. Your sample size doesn’t allow to make a conclusion about this; IgG remained constant (only due to a14-days period?). L332: Also, for day 14? And for both CBD and CBO groups?
Response: We appreciate the reviewer’s observation. However, the observation of a 40–50% decrease in IgM that you mentioned appears to correspond to the consistently lower IgM levels observed in the CBDG group across all timepoints when compare with other treatments (CON and CBDO). There is no decrease in IgM values within the same treatment across all timepoints. This pattern reflects baseline variation rather than treatment or time effects, as confirmed by the absence of statistical significance across groups and timepoints.
L342: You can report data about cortisol levels for both treated groups.
Response: Revised as suggested in Line 365-368.
L363: These advantages were not studied here. So please add references or clearly report that is according to the literature.
Response: Revised as suggested to clarify that these statements are based on existing literature in Line 410.
L369-372: You reports several times the term “notably”. Due to the lack of statistical power, you compare absolute values. So, 1) you make a mention to this problem for each specific comparison (reporting that a new study is required to confirm this hypothesis); 2) do not compare absolute values, or, 3) in this case, it is possible to use the PCA outcomes. Also, you use the term “Trend” other is not a statistical tendency) instead of “pattern).
Response: We have revised the manuscript to (1) replace “trend” with “pattern” where appropriate, (2) avoid overinterpretation of numerical differences that were not statistically significant, and (3) clarify that these observations are hypothesis-generating and should be confirmed in future studies. (Line 421, 450)
L398: Was apparently well tolerated in shelter dogs, no?
Response: Thank you for the observation. We agree that, due to the preliminary nature of the study and the small sample size, the phrasing should be more cautious. We have revised the sentence to reflect this appropriately in the Conclusion.
L400: It is associated with ... (according to the PCA)
Response: Revised as suggested.
L401-402: Please see comment on L274.
Response: Revised as suggested.

Reviewer 3 Report
Comments and Suggestions for Authors
Dear authors,
The article is interesting and provides valuable data about stress-relieving potential of gel-based CBD and CBD oil in dogs. I only have some minor considerations (see the comments below).
Title
From my perspective, the title of the paper does not clearly indicate the research's aims. I suggest you change it to make it clearer and more attractive for readers. For example, something similar to this - Exploring the Stress-Relieving Potential of gel-based CBD and CBD oil in Dogs.
Materials and methods
Please provide the names of the apparatus for hematological, biochemical analysis as well as for cortisol and IgG and IgM.
Results
Considering that immunoglobulins are glycoproteins, I think it would be more appropriate to place the Ig results in table 2.
Line 270 - To avoid repeating the word support(ed), please replace it.
Author Response
The article is interesting and provides valuable data about stress-relieving potential of gel-based CBD and CBD oil in dogs. I only have some minor considerations (see the comments below).
Response : We sincerely appreciate the reviewer’s positive feedback and their recognition of the improvements made in the manuscript. We hope that the revised manuscript will meet your expectations. Our responses to each comment are listed below.
Title : From my perspective, the title of the paper does not clearly indicate the research's aims. I suggest you change it to make it clearer and more attractive for readers. For example, something similar to this - Exploring the Stress-Relieving Potential of gel-based CBD and CBD oil in Dogs.
Response: We thank the reviewer for the helpful suggestion regarding the manuscript title. We agree that highlighting the comparison between gel-based and oil-based CBD enhances the clarity and appeal of the title. Accordingly, we have revised the title to: “A Preliminary Evaluation of the Comparative Efficacy of Gel-Based and Oil-Based CBD on Hematologic and Biochemical Responses in Dogs.”
Materials and methods Please provide the names of the apparatus for hematological, biochemical analysis as well as for cortisol and IgG and IgM.
Response: Revised as suggested in 2.4 Blood Collections and Analyses.
Results Considering that immunoglobulins are glycoproteins, I think it would be more appropriate to place the Ig results in table 2.
Response: Thank you for the suggestion. We agree with your rationale and have moved the immunoglobulin (IgG and IgM) results to Table 2, alongside other serum-derived parameters. The table title and caption have been updated accordingly.
Line 270 - To avoid repeating the word support(ed), please replace it.
Response: Revised as suggested.

Reviewer 4 Report
Comments and Suggestions for Authors
Dear authors,
I have carefully reviewed your manuscript “A Preliminary Evaluation of the Efficacy of Gel-Based CBD on hematologic and Biochemical Responses in Dogs”.
I think that this experiment is nicely designed, with a clear aim and a concise focus on its findings. The results are well presented and the discussion is brief but clear, well-structured and enclosed to the findings reported.
I think that the manuscript is well written and have only some minor comments (see below).
First, the number of animals (n) is quite small in each group. Nonetheless, the authors clearly exposed this as a shortcoming and, indeed, this is common in many research focusing on CBD.
Line 96.- “twelve adult mixed-breed dogs of unknown ages, with an average body”. Although later (in the groups), the authors talked about 2 males and 2 females, I think it could be adequate to describe the composition of the overall population here (sex).
Line 104.- Could the authors provide a concise summary on the tests performed to secure that these animals were healthy?
Line 119.- Is this dose based on any previous report?
Line 156.- Why did the authors did not perform hematology at hour 0?
Line 214.- “No adverse hematological effects, such as anemia..” I would suggest to avoid here the term anemia (these dogs showed low Hb and HCT consistently -below normal levels for dogs-; thus -although not due to the experiment- they were indeed anemic).
Author Response
First, the number of animals (n) is quite small in each group. Nonetheless, the authors clearly exposed this as a shortcoming and, indeed, this is common in many research focusing on CBD.
Line 96.- “twelve adult mixed-breed dogs of unknown ages, with an average body”. Although later (in the groups), the authors talked about 2 males and 2 females, I think it could be adequate to describe the composition of the overall population here (sex).
Response: We thank the reviewer for this helpful suggestion. We have revised the manuscript to include the sex distribution of the overall study population in the Animals and Housing section.
Line 104.- Could the authors provide a concise summary on the tests performed to secure that these animals were healthy?
Response: We appreciate the reviewer’s concern. As noted in the Animals and Housing section, all dogs underwent a veterinary health assessment prior to enrollment. This included clinical examination to confirm the absence of severe clinical illness—defined as persistent diarrhea, inappetence, lethargy, signs of pain, or visible signs consistent with neoplastic disease. No laboratory or imaging diagnostics were performed, given the field conditions and the study’s preliminary design. The inclusion criteria were based on observable physical signs and veterinary judgment to ensure the animals were stable for participation.
Line 119.- Is this dose based on any previous report?
Response: The CBD dose of 4 mg/kg body weight per day was selected based on previous studies demonstrating its safety and efficacy in dogs. Regarding the sampling interval, we chose to assess outcomes every 7 days (Days 1, 7, and 14) to monitor both short- and medium-term effects of CBD supplementation. This interval allowed for the detection of physiological changes while minimizing handling-related stress in the animals. Weekly or biweekly evaluations have also been used in previous CBD studies involving companion animals. We have revised in Discussion Line 320-323.
Line 156.- Why did the authors did not perform hematology at hour 0?
Response: We appreciate the reviewer’s observation. We have 2 main reasons for this matter. Firstly, hematological analysis was performed only at Hour 2 due to ethical and physiological limitations on the total blood volume that could be safely collected from each dog at a single timepoint. In this study, secondly, our primary focus for evaluating acute stress response was based on serum cortisol, which was measured at both Hour 0 and Hour 2. In contrast, hematological parameters were assessed at Hour 2 to provide information on the dogs’ systemic physiological status following treatment and confinement, rather than to detect rapid changes over the short-term. Future studies with larger animals or extended protocols may allow for paired hematological sampling to investigate acute hematologic fluctuations in greater detail.
Line 214.- “No adverse hematological effects, such as anemia..” I would suggest to avoid here the term anemia (these dogs showed low Hb and HCT consistently -below normal levels for dogs-; thus -although not due to the experiment- they were indeed anemic).
Response: We appreciate the reviewer’s careful observation. We agree that the consistently low hemoglobin and hematocrit levels observed across all groups—although present prior to the intervention—could technically reflect an anemic status. To avoid misinterpretation, we have revised the sentence to remove the term "anemia" and instead clarify that no further hematological abnormalities or worsening trends were observed during the study period. This change better reflects the baseline status of the dogs and the focus on treatment-related effects. We have revised in Line 222-223.

Round 2
Reviewer 2 Report
Comments and Suggestions for Authors
Dear authors,
Thank you for providing this revised version. All the comments and suggestions made by this reviewer were taken in consideration, and most of them inserted on this version.